# A Novel Immune Gene-Related Prognostic Score Predicts Survival and Immunotherapy Response in Glioma

**DOI:** 10.3390/medicina59010023

**Published:** 2022-12-22

**Authors:** Xuehui Luo, Qi Wang, Hanmin Tang, Yuetong Chen, Xinyue Li, Jie Chen, Xinyue Zhang, Yuesen Li, Jiahao Sun, Suxia Han

**Affiliations:** Department of Radiation Oncology, First Affiliated Hospital of Xi’an Jiaotong University, Xi’an 710049, China

**Keywords:** RNA sequencing, immune-related gene, prognostic biomarker, glioma, immune checkpoint blockade, tumor immune microenvironment, WGCNA

## Abstract

*Background and Objectives:* The clinical prognosis and survival prediction of glioma based on gene signatures derived from heterogeneous tumor cells are unsatisfactory. This study aimed to construct an immune gene-related prognostic score model to predict the prognosis of glioma and identify patients who may benefit from immunotherapy. *Methods:* 23 immune-related genes (IRGs) associated with glioma prognosis were identified through weighted gene co-expression network analysis (WGCNA) and Univariate Cox regression analysis based on large-scale RNA-seq data. Eight IRGs were retained as candidate predictors and formed an immune gene-related prognostic score (IGRPS) by multifactorial Cox regression analysis. The potential efficacy of immune checkpoint blockade (ICB) therapy of different subgroups was compared by The Tumor Immune Dysfunction and Exclusion (TIDE) algorithm. We further adopted a series of bioinformatic methods to characterize the differences in clinicopathological features and the immune microenvironment between the different risk groups. Finally, a nomogram integrating IGRPS and clinicopathological characteristics was built to accurately predict the prognosis of glioma. *Results:* Patients in the low-risk group had a better prognosis than those in the high-risk group. Patients in the high-risk group showed higher TIDE scores and poorer responses to ICB therapy, while patients in the low-risk group may benefit more from ICB therapy. The distribution of age and tumor grade between the two subgroups was significantly different. Patients with low IGRPS harbor a high proportion of natural killer cells and are sensitive to ICB treatment. While patients with high IGRPS display relatively poor prognosis, a higher expression level of DNA mismatch repair genes, high infiltrating of immunosuppressive cells, and poor ICB therapeutic outcomes. *Conclusions:* We demonstrated that the IGRPS model can independently predict the clinical prognosis as well as the ICB therapy responses of glioma patients, thus having important implications on the design of immune-based therapeutic strategies.

## 1. Introduction

Glioma is the most common primary intracranial tumor, accounting for 81% of all primary malignant brain tumors [1]. The overall incidence of gliomas is about six cases per 100,000 population and nearly half of the cases are very aggressive glioblastomas, for which the 5-year survival rate is just 5% [2]. Despite improvements to existing treatment methods, such as surgery, chemotherapy, and radiotherapy, the prognosis of patients with malignant glioma is still very poor [3]. Reasons for this include the growth of invasive tumors without clear margins, lesions located in important areas of the brain, and inoperable lesions, all of which can lead to post-operative recurrence; the blood–brain barrier impedes drug delivery, leading to chemotherapeutic resistance [4]; due to the rapid progression of gliomas, radiotherapy techniques do not significantly improve the overall survival (OS) of glioma patients; the complex heterogeneity of gliomas produces unsatisfactory results of molecular targeted therapy. All of this indicates that existing treatments are not effective in improving the prognosis of gliomas and new treatment strategies are urgently needed.

Immunotherapy has brought significant survival benefits in the treatment of many tumors [5,6,7]. A variety of glioma immunotherapies have been developed: 1. Chimeric antigen receptor T cell immunotherapy can target tumor surface antigens and stimulate the immune system to kill tumor cells by collecting autologous T cells targeting tumor antigens and genetic engineering; 2. Oncolytic virus therapy activates the anti-tumor immune response by infecting and lysing tumor cells; 3. Dendritic cell vaccine can expand and activate dendritic cells carrying tumor antigens, inducing the activation of cytotoxic T lymphocytes; 4. Glioma cells typically evade immune surveillance by activating immune checkpoint ligands such as Programmed Death Receptor1 (PD-1). Immune checkpoint blockade (ICB) can make T cells continuously activated to eliminate tumor cells [8,9,10]. These methods have shown therapeutic potential in specific glioma patients [11]. However, due to the tumor heterogeneity of gliomas, the prognosis and immunotherapeutic response of glioma patients vary widely among individuals. Phase II trials of ICB treatment for gliomas have not produced consistent results [12] and the efficacy of immunotherapies cannot be confirmed in larger randomized trials. A more accurate definition of target groups with positive immune responses would be a breakthrough. A more accurate classification of gliomas based on effective evaluation indicators of individual patients would enable immunotherapy to achieve stable and positive efficacy in screened glioma patients. However, there is still a lack of validated biomarkers for gliomas to forecast survival and potential immunotherapy benefit. Therefore, a novel biomarker for predicting the OS of gliomas and identifying target patients suitable for ICB therapy is urgently needed.

The tumor immune microenvironment (TIME) consists of multitudinous important inflammatory infiltrating cells, complex inflammatory signaling molecules and an extracellular matrix [13], which is crucial to the tumorigenesis and metastasis of glioma. Multiple immunosuppressive cells and soluble characteristic factors and their interactions, as well as multiple immunosuppressive mechanisms in TIME, were found to inhibit antitumor immunity [14], diminishing the efficacy of immunotherapy. The up-regulation and activation of important immune-related genes (IRGs) in TIME were shown to be essential to the regulation and reconstruction of the immune system [15]. Immune gene expression signatures were used to characterize the immunophenotyping of patients across multiple tumor types in previous studies [16,17,18]. The immune gene expression profile of tumors varies by molecular typing and has prognostic value [19,20]. Prognostic prediction models based on the relationship between immune gene expression profiles and tumor survival prognosis have been developed in osteosarcoma, lung adenocarcinoma, colon cancer, etc. [21,22,23]. An in-depth search for IRGs and exploration of the TIME in gliomas may contribute to understanding the heterogeneity of gliomas and improving the accuracy of predicting prognosis and response to immunotherapy in glioma patients. However, the association between IRGs, prognosis, immunotherapeutic responses and TIME has not been well characterized in gliomas. To date, no accurate risk score based on IRGs has been developed for prognostic prediction and selection of ICB treatment in glioma.

In this study, we first analyzed the RNA-sequencing data of glioma samples and developed a novel immune gene-related prognostic score (IGRPS) using a series of bioinformatics algorithms including WGCNA and Cox regression analysis. Second, we compared the potential ICB therapeutic benefits of IGRPS-defined subgroups using the Tumor Immune Dysfunction and Exclusion (TIDE) algorithm. Thirdly, we delved into the correlation between IGRPS and the clinicopathological characteristics (e.g., age, gender, tumor grade, survival status, survival time, etc.) of risk in glioma patients. Finally, we constructed a nomogram to accurately predict the prognosis of gliomas (the flow chart is shown in Figure 1).

## 2. Materials and Methods

### 2.1. Patients and Datasets

We downloaded the mRNA expression sequencing data and clinical information of gliomas from the Cancer Genome Atlas (TCGA) database (http://cancergenome.nih.gov/abouttcga, accessed on 5 January 2022) and obtained data of normal brain tissues from the Genotype-Tissue Expression (GTEx) database (http://xena.ucsc.edu/, accessed on 5 January 2022). Samples with incomplete survival data were eliminated and 670 glioma patients with complete prognostic information were enrolled in the subsequent analysis (the baseline clinical data are summarized in Table A1). The mRNA sequencing data and corresponding clinical parameters of the validation set were collected from the Chinese Glioma Genome Atlas (CGGA) database (http://www.cgga.org.cn/, accessed on 5 January 2022). IRGs used in this study were obtained from the ImmPort (https://www.immport.org/, accessed on 7 January 2022) and InnateDB (https://innatedb.com, accessed on 7 January 2022) databases.

### 2.2. Screening of Immune-Related Differentially Expressed mRNAs in Glioma Patients

The differentially expressed genes (DEGs) between TCGA glioma patients (*n* = 670) and GTEx healthy brain samples (*n* = 265) were screened by limma package and Wilcox test [24] (*p*-value < 0.05, log2FC > 1) and then cross-extracted with the IRGs to identify immune-related DEGs. Gene Ontology (GO) analysis [25] and Kyoto Encyclopedia of Genes and Genomes (KEGG) analysis [26] were performed on immune-related DEGs to explore potential molecular mechanisms via the clusterProfiler package of R.

### 2.3. Identification of Prognostic Immune-Related Hub Genes

Weighted gene co-expression network analysis (WGCNA) weights the similarity between genes to build a scale-free network subjected to power-law distribution, in which a few highly connected genes (hub genes) are considered vital and may be candidate biomarkers [27].

We used the WGCNA package to analyze 785 immune-related DEGs. First, we calculated the Pearson correlation between genes and established a similarity matrix. Second, we used the pickSoftThreshold function to obtain the optimal power of weighted adjacency function parameters, also called the soft threshold. The topological overlap matrix was obtained by quantitative analysis of the weighted node similarity. Next, we used the hierarchical clustering function to subdivide modules, and then calculated the module eigengene and module significance according to the correlation between module genes and the clinical phenotype “glioma with or without.” The brown module, which showed the highest significance, was deemed to be associated with the external clinical phenotype. Therefore, 74 genes in the brown module were analyzed through the STRING online database (http://www.string-db.org/, accessed on 7 February 2022) and processed by Cytoscape software (download from http://www.cytoscape.org/, version: 3.9.0, accessed on 7 February 2022) [28,29]. Genes ranked top 40 were the hub genes. In total, 23 hub genes related to prognosis were identified by survminer and survival packages in R (HR > 1; *p* < 0.05) [30].

### 2.4. Construction and Validation of the Immune-Gene-Related Prognostic Score

Multivariate COX regression analysis was performed on 23 prognostic hub genes. We identified eight IRGs that independently affected the OS of glioma patients and constructed an immune-gene-related prognostic score (IGRPS). The IGRPS of each sample was obtained by calculating the product of gene expression of these eight characteristic IRGs and their coefficient from Cox model and then summing them up sequentially. Glioma patients were subdivided into different subgroups according to the median value of IGRPS. The prognostic ability of IGRPS was assessed via univariate analysis, using 325 samples from the CGGA cohort to validate.

### 2.5. Exploration of Tumor Immune Microenvironment in Different IGRPS Subgroups

Gene set enrichment analysis (GSEA) [31] was adopted to raise differential expression of biologically relevant pathways. We used reference gene set file (c2.cp.kegg.v7.4.symbols.gmt) and clusterProfiler package to perform GSEA on different IGRPS subgroups and visualized the important enrichment pathways using the enrichplotsoftware package. CIBERSORT is a deconvolution method for gene expression microarray data to characterize immune cell infiltration [32,33]. We adopted CIBERSORT to determine the relative amount of 22 immune cells in different risk subgroups, and grouped patients into high and low immune cell infiltration subgroups. Survival analysis was further utilized to investigate the association between immune cell infiltration degree and prognosis of gliomas. Single-sample gene set enrichment analysis (ssGSEA) [34,35] was adopted to characterize the enrichment levels of immune functional features [36]. We used gene set variation analysis (GSVA) software package [37] to perform ssGSEA analysis in the two subgroups and explored the enrichment abundance of 29 immune-related functional features and also probed the association among immune functional features and prognosis of gliomas using survival analysis.

### 2.6. The Contrast of the Clinicopathological Features in Different IGRPS Subgroups

We analyzed the distribution of patient status and clinicopathological features in different IGRPS subgroups. The distribution of age, gender, and tumor grade of patients in the different risk subgroups were compared by Chi-square test to explore the relationship between IGRPS and clinicopathological characteristics of glioma patients.

### 2.7. Prediction of the Benefits of Immune Checkpoint Blockade Therapy in Different IGRPS Subgroups

The TIDE algorithm [38] integrates the gene expression characteristics of T cell dysfunction and T cell exclusion for predicting the potential clinical responses to ICB therapy [39]. High TIDE scores indicate poor efficacy of ICB treatment. We adopted the TIDE algorithm to compare potential immunotherapy benefits in different risk groups. We further compared the expression of DNA mismatch repair (MMR) genes [40] in high- and low-risk groups. The deficiency or inactivation of MMR genes means mismatch repair defect, which is partly equivalent to high-frequency microsatellite instability (MSI-H) [41]. MSI-H usually leads to a series of molecular and biological changes, therefore tumors with MSI-H phenotype have a higher objective response rate to ICB therapy [42].

### 2.8. Construction of the Nomogram

We built a ROC curve to predict the 1-, 2-, and 3-year survival rates of glioma patients to evaluate the prognostic accuracy of IGRPS. We further compared the prediction capabilities of IGRPS with TIDE and TIS by the decision curve analysis. Finally, we used the rms and survival packages in R to construct a nomogram by integrating risk score, age, and tumor grade to predict the OS of glioma patients.

### 2.9. Statistical Analysis

R software (downloaded from https://www.r-project.org/, version 4.1.1, accessed on 7 February 2022) and SPSS software (download at https://www.ibm.com/spss, version 25.0, accessed on 7 February 2022) were used for statistical analysis. The correlation between continuous variables and classified variables was analyzed by *t*-test and Chi-square test, respectively. Wilcoxon test was used to compare TIDE scores among groups. Kaplan–Meier method was used to calculate the survival rate, Log rank test was used to test the difference in survival rate, and Cox regression method was used for multivariate analysis. Bilateral *p* < 0.05 was considered statistically significant.

## 3. Results

### 3.1. Identification of Prognostic Immune-Related mRNAs

We obtained 12,071 DEGs through differential analysis, of which 7181 genes were up-regulated and 4890 genes were down-regulated in tumor tissues, as shown in the volcanic map (Figure 2A). In total, 785 IRDEGs were further identified by integrating IRGs with DEGs (Figure 2B). The annotation results of GO and KEGG functional enrichment analysis on the IRDEGs are shown in Figure A1. We performed WGCNA on IRDEGs and chose the optimal soft threshold β = 5 to structure a gene co-expression network (scale-free topological fit index reached 0.90) (Figure 2C). The highly connected genes were identified into modules by hierarchical clustering, resulting in a dendrogram of genes (Figure 2D). Next, we identified four modules and the brown module was closely connected with gliomas (Figure 2E). Therefore, PPI network analysis and visualization were performed on 74 genes of the brown module (Figure 2F). According to the calculation and ranking of internode connectivity, we selected the top 40 IRGs as hub genes. Finally, 23 immune-related hub genes that were significantly associated with the OS of glioma patients were identified (Figure A2). The first 10 Go and KEGG enrichment pathways of 74 genes in the brown module are shown in Figure A3.

### 3.2. Survival Prognosis in Different IGRPS Subgroups

We performed an independent prognostic analysis of 23 prognostic IRGs using multivariate Cox regression analysis (Figure 3A). Eight genes were recognized as independent prognostic factors for gliomas (Figure 3B). High-risk genes are marked as red (HR > 1) in the forest map, while low-risk genes are marked as green. Using a risk calculation formula incorporating these eight genes, IGRPS = FGFR1×(0.3818)+ FLT3×(−1.0789)+ VTN×(−0.4290)+ NR2C1×(−0.4888)+ SEMA4G×(−1.4351)+ CFP×(0.6849)+ S100P×(−0.2137)+ CHGB×(−0.1426), we calculated the IGRPS for each glioma patient and successfully divided them into high- and low-risk subgroups by IGRPS. Principal component analysis showed that the high- and low-risk subgroups had a good degree of differentiation based on the eight signature IRGs, (Figure 3C,D). Patients with low-IGRPS risk showed better survival outcomes (*p* < 0.001) (Figure 3E). The CGGA data set (*n* = 325) was utilized to verify the predictive effectiveness of the IGRPs and reached the same conclusion (Figure 3F).

### 3.3. Landscapes of the Tumor Immune Microenvironment in Different IGRPS Subgroups

GSEA showed that the immune-related DEGs enriched in the high-IGRPS subgroup were associated with cell cycle, cytokine–cytokine receptor interactions, and other related pathways (Figure 4A). While the top three potential molecular pathways enriched in the low-IGRPS subgroup were the calcium signaling pathway, long-term inhibition, and long-term potentiation (Figure 4B). Next, we obtained the relative infiltration percentages of immune cells of glioma patients through the CIBERSORT algorithm (Figure A4). The distribution of 15 immune cells was statistically different between the two subgroups (*p* < 0.05). Specifically, M2 and M0 macrophages and neutrophils showed higher infiltration abundance in the high-IGRPS subgroup, while monocytes and activated hypertrophy cells were more abundant in the low-IGRPS subgroup (Figure 4C). The box plot showed that among the 28 immune functional features with significant distribution differences (*p* < 0.05), except NK cells, other immune function features, such as B cells and Tregs showed higher enrichment abundance in the high-IGRPS subgroup (Figure 4D). Survival analysis showed that patients with a high infiltration degree of macrophages or CD8 + T cells had a worse prognosis, while patients with high infiltration of monocytes showed a better survival outcome (Figure 4E,F and Figure A5). Higher enrichment levels of Treg cells, cytolytic activity, and checkpoints were related to poor survival prognosis, whereas patients with high NK cell infiltration had better prognoses (Figure 4G,H and Figure A6).

### 3.4. Characteristics of Clinicopathological Features in Different IGRPS Subgroups

This study found that mortality was significantly higher in the high-IGRPS subgroup (Figure 5A). The expression profiles of eight prognostic IRGs in the two subgroups are shown in the heatmap (Figure 5B). The clinicopathological characteristics of 670 glioma patients in the different risk subgroups from the TCGA cohort are shown in Figure 5C, and those from the CGGA cohort are shown in Figure 5D. The distribution of the age and tumor grade between the two risk subgroups were significantly different but not the gender (Table A2).

### 3.5. Patients with Low IGRPS May Be a Higher Priority for ICB Treatment

The results of TIDE showed that patients in the high-IGRPS subgroup had higher TIDE scores, which was associated with a poor response to ICB treatment, and those in the low-IGRPS subgroup were higher priorities for ICB treatment (Figure 6A). The violin plots for immune exclusion and dysfunction are shown in Figure 6B,C. Analysis of four major MMR genes showed that MSH2, MSH6, and MLH1 were remarkably highly expressed in the high-IGRPS group (Figure 6C–E). PMS2 was highly expressed in the low-IGRPS group with an inconspicuous difference (Figure A7). In our study, the lower expression of MMR genes in the low-IGRPS group indicated that patients in the low-risk group had a better response rate to ICB treatment, which was consistent with the better potential ICB therapeutic benefit for low-risk patients predicted by the TIDE algorithm.

### 3.6. Prognostic Value of IGRPS

We validated the prognostic ability of IGRPS using ROC curves. The AUCs of 1-, 2-, and 3-year survival outcomes were 0.868, 0.895, and 0.889, which indicated the good prognostic ability of IGRPS (Figure 7A). We further compared the predictive value of IGRPS to those of TIDE and TIS with respect to predicting the 3-year OS of glioma patients. The AUCs were 0.889, 0.538, and 0.783, and IGRPS showed significantly better accuracy (Figure 7B). Univariate and multivariate independent prognostic analyses indicated that IGRPS, age, and tumor grade were independent prognostic factors for gliomas (Figure 7C,D). Finally, we integrated IGRPS, age, and tumor grade to form a nomogram to accurately forecast the OS of gliomas (Figure 7E). The calibration curve [22] showed considerable goodness of fit with the ideal model (Figure 7F).

## 4. Discussion

According to the traditional World Health Organization (WHO) classification, gliomas are divided into grades I to IV based on tumor invasiveness [43], with the survival and treatment of gliomas varying with grade. However, due to the complex heterogeneity of gliomas, individual differences in prognosis and treatment response still exist even among patients with the same subtype of glioma, posing a great challenge to clinical decision-making in gliomas [44,45]. There is a need to develop a new prognostic score for gliomas based on the highly heterogeneous biological characteristics of the individual patient to accurately predict the prognosis and immune response to gliomas.

In the study, we performed a high-resolution differential analysis on glioma RNA-seq data [46] and deeply studied the IRG expression profiles of glioma patients using WGCNA analysis. We developed IGRPS, which can serve as an independent prognostic indicator for glioma. After stratification with IGRPS, patients in different subgroups showed significant differences in prognosis and immune response. Specifically, patients in the low-IGRPS subgroup had better outcomes and were more suitable for ICB therapy. ROC curve analysis further confirmed that IGRPS had excellent prediction accuracy. This suggests that IGRPS can be used to optimize the existing classification of gliomas and improve the accuracy of prognosis predictions of gliomas. We integrated IGRPS, age, and tumor grade to form a nomogram for predicting the outcomes of gliomas to help clinicians better choose different therapies for patients with different risk scores, improving their prognosis and quality of life.

IGRPS consists of eight genes (FGFR1, FLT3, VTN, NR2C1, SEMA4G, CFP, S100P, CHGB). In previous studies, dysregulation of FGFR1 signaling was related to tumor progression and tumor immune therapeutic resistance [47]. The expression of FGFR1 was higher in tumors with high malignancy [48]. FGFR1 was defined as a high-risk gene linked to poor prognosis in this study (Figure 5B). FGFR1 is used as a molecular target for antitumor therapy. Several FGFR1 inhibitors were developed for cancer treatment [49]. In gliomas, inactivation of the PI3K/AKT pathway via targeted FGFR1 inhibition increased the sensitivity of patients to temozolomide [50]. FLT3 (receptor-type tyrosine-protein kinase) mutations were related to poor prognosis in gliomas [51]. In our study, FLT3 was a low-risk gene that was highly expressed in the low-IGRPS subgroup with better outcomes. VTN (vitronectin) is an adhesive glycoprotein, which can mediate tumor progression and migration by promoting angiogenesis [52]. The high expression of VTN was related to poor prognosis in melanoma patients [53]. VTN is one of the main factors that induces tumor migration in glioma [54]. S100P (nuclear autoantigen Sp100) can control cell growth and differentiation. In previous studies, the inducing overexpression of SP100 reduced the proliferation and migration of glioblastoma tumor cells [55], which indicated that SP100 might be a promising target for glioma. The role of other genes (NR2C1, SEMA4G, CFP, CHGB) included in IGRPS has not been elucidated in gliomas and other tumors. Further studies are needed to investigate their molecular characteristics and tumor-related functions, which might bring new insights for anti-tumor therapeutic targets.

GO enrichment function of DEGs and hub genes both pointed to immune activities, especially, the interaction events between immune cells and cytokines. KEGG analysis showed that the MAPK signaling pathway was the most significant pathway. Further analysis showed that the cell cycle and cytokine receptor interactions were significantly concentrated in the high-IGRPS subgroup. These potential molecular signatures mined based on differentially expressed genes and risk subgroup profiles have stimulated our interest in the tumor immune microenvironment, which consists of a variety of cells and components such as immune cells, stromal cells, and secretory molecules [56,57].

We further evaluated the landscape of TIME in different IGRPS subgroups. The results showed that patients in the high-IGRPS group were enriched with more immunosuppressive cells, such as M0 and M2 macrophages and neutrophils, whereas monocytes and activated NK cells had a higher infiltration in the low-IGRPS subgroup. The heterogeneity of tumor-infiltrating immune cell infiltration in different IGRPS subgroups may be the key to glioma malignant development and immunotherapy failure. In previous studies, macrophages were identified as a negative prognostic factor in glioma patients [58]. Tumor-derived chemokines recruit monocytes into TIME and further differentiate into tumor-associated macrophages (TAM), which promotes tumor cell proliferation and induces angiogenesis [59]. Blocking tumor progression by targeting TAMs in gliomas is a promising approach [60]. In other studies, high resting memory CD4-T cell infiltration was a positive prognostic marker in head and neck squamous cell carcinoma [61]. In our study, ssGSEA also revealed the differential enrichment level of immune function features in different IGRPS subgroups. The NK cell abundance was higher in the low-IGRPS group. Tregs showed a higher enrichment level in the high-IGRPS group. Highly expressed NK cells were associated with a positive prognosis, whereas patients with higher Treg cell enrichment levels showed a bleak prognosis. The unique advantages of NK cells as potential therapeutic targets for glioma were widely explored [62,63]. Immunotherapy targeting NK cells is promising for glioma treatment [64]. Tregs were found to be associated with the immunosuppression of tumors in previous studies [65,66]. Negative regulation of the immunosuppressive effects of Tregs by blocking their recruitment to TIME may be a promising therapeutic strategy for glioma [67]. These results suggest that glioma patients stratified by IGRPS have heterogeneous TIME, in which the heterogeneous infiltration of macrophages, NK cells, Treg cells, and other important immune cells significantly affects the prognosis and treatment response of glioma patients, providing new insights for potential glioma immunotherapeutic targets.

### 4.1. Limitations

The ability of IGRPS model to predict immune response for glioma is indirectly estimated by the TIDE algorithm. There is a lack of a cohort of glioma patients who have received immunotherapy to directly validate the predictive value of the model for immune response. Immunotherapy has not been approved for glioma and glioma immunotherapy strategies are still in the pre-clinical and clinical research stages. The completed or ongoing clinical studies of glioma ICB immunotherapy are currently in phase I/II clinical studies [8], where the survival prognosis of glioma patients who have received ICB immunotherapy is difficult to obtain. Therefore, we did not take the parameter of receiving immunotherapy as the inclusion criteria. We hope to obtain enough samples of glioma patients who have received immunotherapy, so as to directly verify the model in the future. In addition, due to the limited clinical features of glioma that we can obtain from the TCGA database, the exploration of the relationship between the IGRPS model and clinical characteristics is not sufficient. We hope to further explore the clinical application of this model with more clinical information in the future, such as whether patients have received chemotherapy or radiotherapy or other treatments.

### 4.2. Strengths

The predictive biomarkers of immunotherapy have been extensively studied. At present, the most widely used method is probably to detect PD-L1 immunohistochemistry in tissue samples. Although many studies have reported that patients with high PD-L1 expression have a high response rate to ICB treatment, quantification of critical levels is complex, which limits the utility of this indicator. The TIDE algorithm was developed to predict the potential treatment response of ICB therapy and its predictive accuracy was reported to be superior to PD-L1 expression assay [68,69,70]. However, the calculation of TIDE score based on large-scale high-throughput genomic sequencing data incurs heavier costs and burdens that limit its widespread use in the clinic. This study stratifies glioma patients by IGRPS risk, with low-risk subtypes having lower TIDE scores and being more likely to benefit from ICB therapies, reducing the genes to be sequenced to eight immune-related genes, thus significantly reducing the cost and burden for patients. We hope to filter out glioma patients with good immune responses through IGRPS model so that immunotherapy can obtain stable clinical benefits in such target patients, and thus shorten the distance between clinical research and the clinical application of immunotherapy. It is hoped that the predictive value of immunotherapy efficacy of this model will contribute to the design of clinical immunotherapy research and promote the development of precise immunotherapy for glioma.

## 5. Conclusions

Compared with traditional tumor grade, IGRPS is a more powerful prognostic biomarker, which can be used to optimize the risk stratification for glioma patients. Patients in the low-risk group had a better prognosis. In particular, patients in the low-risk group exhibited superior potential benefits from ICB therapy and IGRPS may be useful for counseling patients on immunotherapy and help guide individualized treatment and precise immunotherapy of glioma. Furthermore, multi-omic analysis conducted on IGRPS-defined subgroups has contributed to the understanding of the heterogeneity of TIIM in glioma. Immune cells such as macrophages, NK cells, and Tregs showed encouraging prognostic correlations. Novel immunotherapies targeting significant immune infiltrating cells in the TIIM are feasible and beneficial, though further studies will be necessary to confirm this.

## Figures and Tables

**Figure 1 medicina-59-00023-f001:**
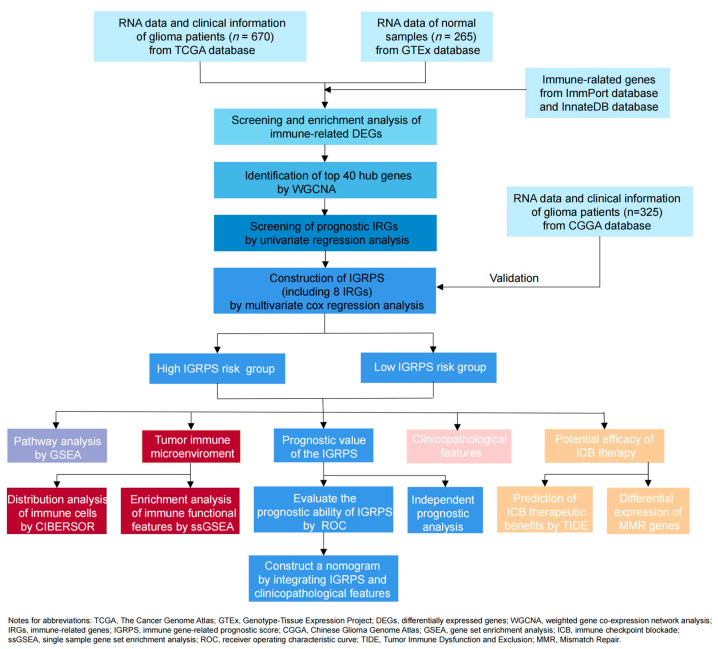
Flow chart.

**Figure 2 medicina-59-00023-f002:**
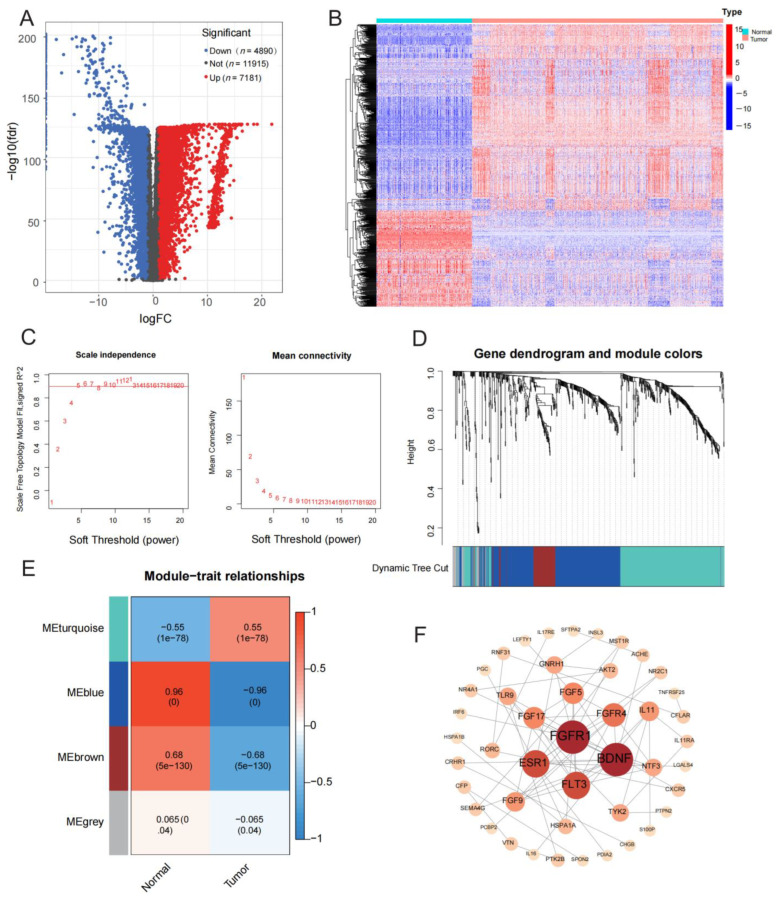
Screening of differentially expressed genes (DEGs) in glioma and identification of immune-related hub genes. (**A**) Volcanic map of DEGs between gliomas and normal samples. (**B**) Heatmap of the immune-related DEGs. (**C**) Determination of the optimal soft threshold. (**D**) Gene dendrogram and clustering modules. (**E**) Correlation and significance between modules and glioma. (**F**) Protein interaction (PPI) network diagram of significant modular genes.

**Figure 3 medicina-59-00023-f003:**
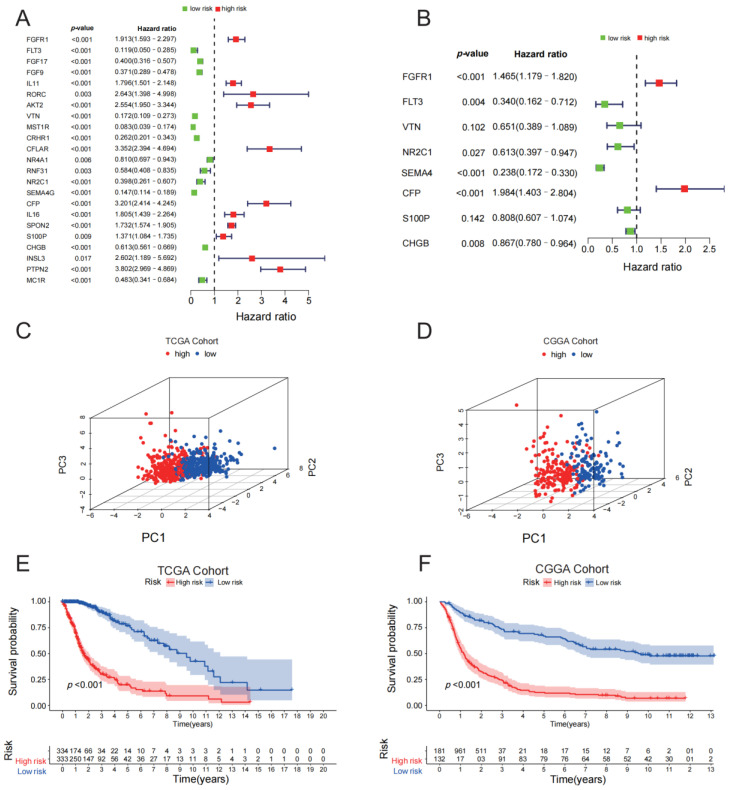
Construction of the immune gene-related prognostic score (IGRPS). (**A**) Univariate Cox analysis of 23 prognostic immune-related hub genes. (**B**) Eight signature IRGs identified by multivariate COX analysis. (**C**,**D**) Principal component analysis of eight signature IRGs. (**E**,**F**) Kaplan–Meier curves of different IGRPS subgroups in two cohorts.

**Figure 4 medicina-59-00023-f004:**
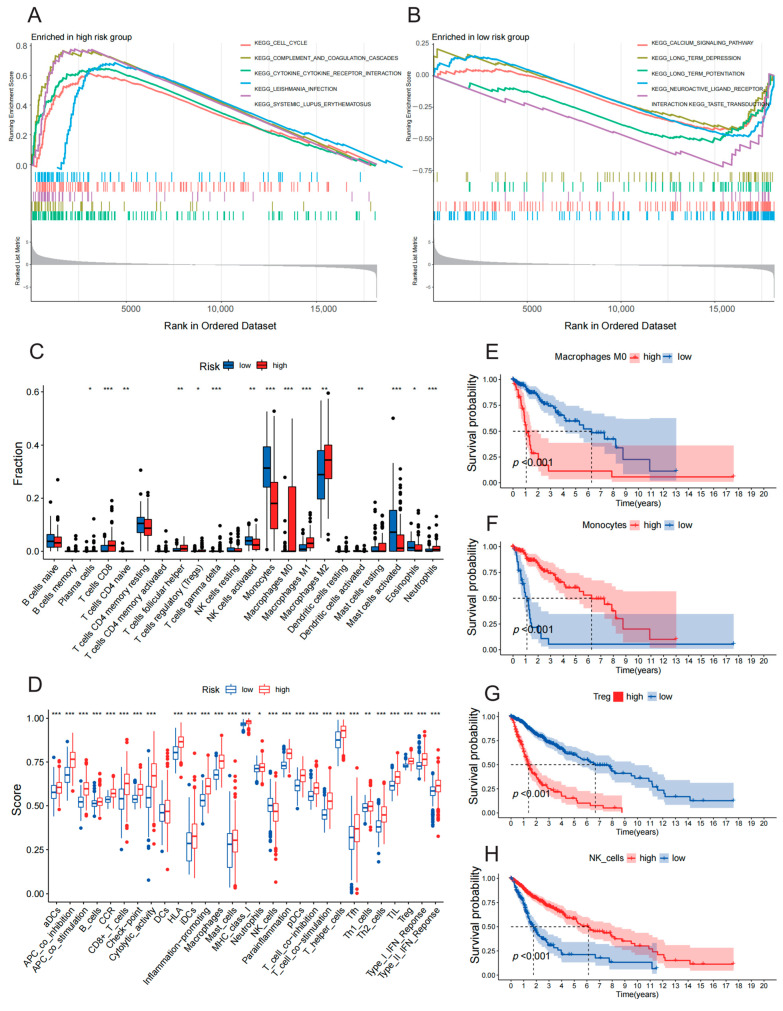
Landscapes of the tumor immune microenvironment. (**A**) Gene sets enriched in the high-IGRPS subgroup. (**B**) Gene sets enriched in the low-IGRPS subgroup. (**C**) The relative infiltration fraction of 22 immune cells in different subgroups. The distribution of 15 immune cells in the two subgroups was statistically different (ns: not significant, * *p* < 0.05, ** *p* < 0.01, *** *p* < 0.001). (**D**) Enrichment abundance of immune function features in different subgroups. (**E**–**H**) Kaplan–Meier curves of M0 macrophage, monocyte, Treg, and NK cell infiltration level.

**Figure 5 medicina-59-00023-f005:**
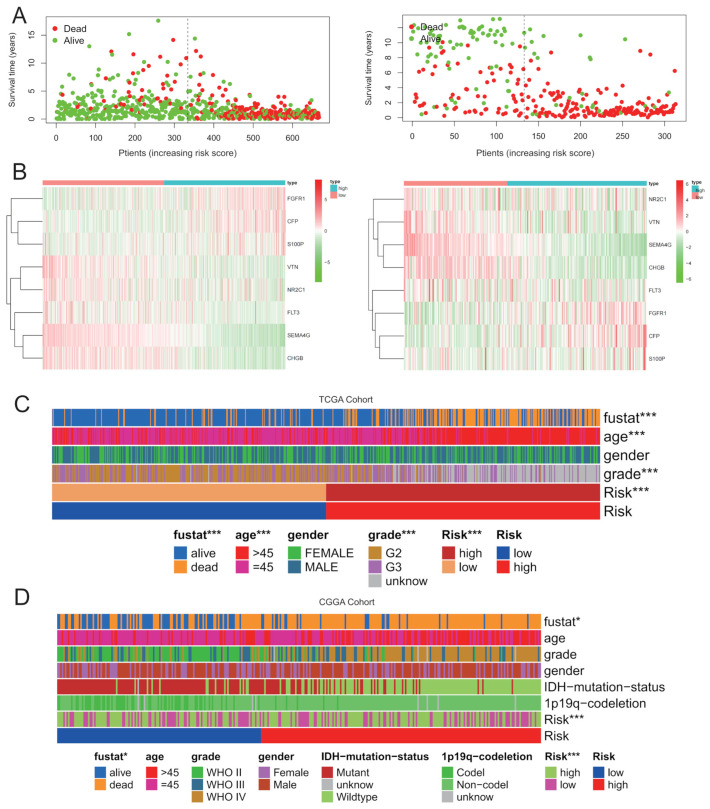
Patient status distribution and clinicopathological features in different IGRPS subgroups. (**A**) Patient status distribution in different IGRPS subgroups. The dots present patient survival status. The X-axis is the number of patients, ranked by increased risk score and the Y-axis is survival rate. (**B**) Heatmap of the expression of eight signature IRGs in patients with different IGRPS risk in two cohorts. (**C**,**D**) Clinicopathological feature heatmaps of IGRPS in the two cohorts. * *p* < 0.05, *** *p* < 0.001

**Figure 6 medicina-59-00023-f006:**
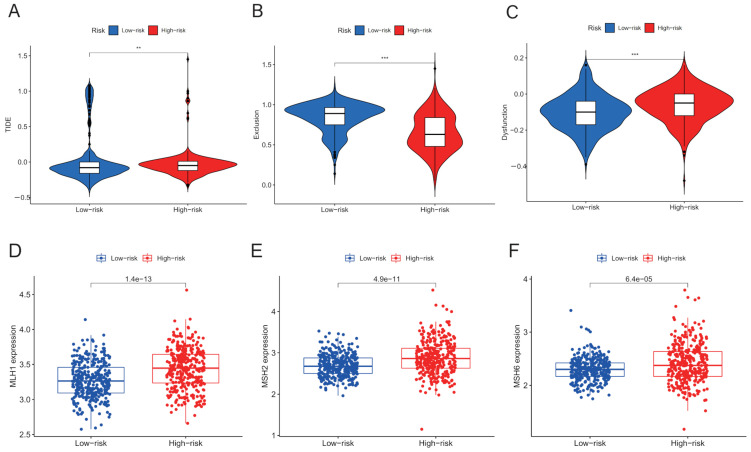
Potential responses to ICB therapy of different IGRPS subgroups. (**A**–**C**) TIDE, T cell exclusion and dysfunction score in different subgroups. Wilcoxon test compared scores between the two groups. (**D**–**F**) Box plots visualizing significantly differentially expressed MMR genes (MLH1, MSH2, and MSH6) in the two groups. ** *p* < 0.01, *** *p* < 0.001

**Figure 7 medicina-59-00023-f007:**
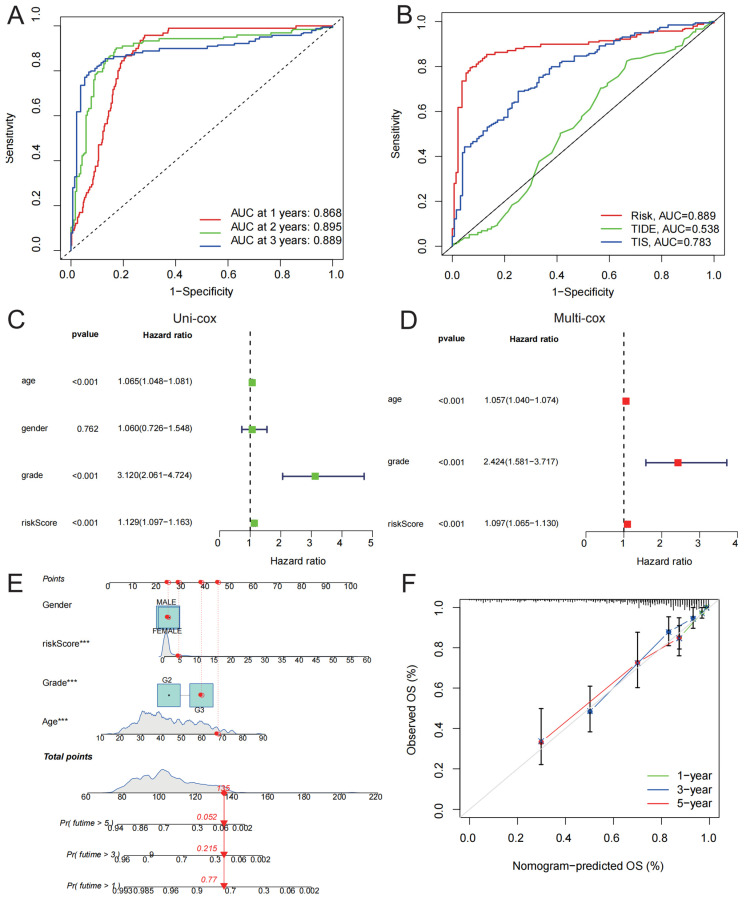
Prognostic value of IGRPS. (**A**) ROC curve for predicting the 1-, 2-, and 3-year survival rates in gliomas. (**B**) ROC curve for predicting the 3-year survival rates in glioma compares predictive power of IGRPS, TIDE, and TIS. (**C**) Univariate independent prognostic analysis of IGRPS. (**D**) Multivariate independent prognostic analysis of IGRPS. (**E**) Nomogram formed by integrating IGRPS and clinicopathological features predicts the glioma prognosis. (**F**) The calibration curve evaluates the accuracy of nomograms for predicting the prognosis of gliomas. *** *p* < 0.001

## Data Availability

The datasets provided in this study are available in online databases and the article contains the login URLs for the databases used.

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
