# Peer review of "A Novel Immune Gene-Related Prognostic Score Predicts Survival and Immunotherapy Response in Glioma"

_medicina, 2022, doi:10.3390/medicina59010023_

Round 1

Reviewer 1 Report

Comments on “A novel immune gene-related prognostic score predicts survival and immunotherapy response in glioma”

The study done by Luo et al. reported a score model with IGRs that can predict the clinical prognosis as well as therapy responses of glioma patients independently. However, the manuscript lacks some important information related to the study and needs extensive revision. Many of the observations are not discussed at all. The comments highlighted below needs to be addressed and revised thoroughly to make it a reliable and meaningful study.

1.      In the abstract, the authors mention that the construction of the immune gene-related prognostic score was established based on the 8 IRG’s, however their study shows that the analysis of 23 prognostic immune-related hub genes is equally important for establishing the prognostic score. Thus, this information needs to be included in the abstract.

2.      Include the full form for NK cells in the abstract.

3.      In the introduction, line 35 the authors introduce OS of glioma patients for the first time. Please include the full form and a brief introduction of OS.

4.      The introduction line 43-44, mention in a bit detail about the glioma immunotherapies that have been developed so far.

5.      In line 64, cite more references since this is one of the major aspects the study is based upon.

6.      In line 76 of introduction, give examples of the clinicopathological characteristics of glioma patients.

7.      There are quite several places where the authors have used abbreviations but not full forms and this is very inconvenient for the readers. Please include a separate list of abbreviations in the manuscript.

8.      In Figure 2 B please redo the labelling’s with larger fonts. Also, in 2D the colours modules in the dynamic tree cut of the dendrogram needs to be explained.

9.      In Figure3 A and B, for better data representations, highlight the genes in the positive hazard ratio range with different colours.

10.   In Figure 4, please use bigger fonts to label the y axis for A and B. Also, explain the observations in detail noted in in E F G and H. Why survival probability of monocytes shows opposite trend to that of the other immune related cells between IGRPS subgroups.

11.   The X axis labelling of Figure 5 A has typo error. Correct it please.

12.   Please club Figure 6 D-F into a single comprehensive figure.

13.   Please make a table for the univariate and multivariate independent prognostic analysis of IGRPS (Figure 7C and D) instead of figures. That way the data would be presented in a more lucid manner.

14.   Is there any standard method for the raw data pre-processing? Any convention or anything else? If yes, please cite and describe the background why it is needed.

15.   In the line 321 in discussion the authors mention that “IGRPS consists of eight genes (FGFR1, FLT3, VTN, NR2C1, SEMA4G, CFP, S100P, CHGB)”, if that is the case then why was the analysis of 23 prognostic immune-related hub genes carried out? Please make sure to discuss about the outcomes and significance of each experiment done in the study in discussion section of the manuscript.

Reviewer 2 Report

The authors did a study about an immune gene-related prognostic score to predict survival and immunotherapy response in glioma. This manuscript is well written and presentation is good. However i have some concerns before pubblication:

1. the conclusion section in the manuscript could be better exhaustive. 

2. the manuscript is hard to read, maybe the collocations of figures should be different and to my opinion the discussione could less long.

Author Response

Dear Editor and the Reviewers,

We are grateful that Medicina us an opportunity to reconsider our revised manuscript titled “A novel immune gene-related prognostic score predicts survival and immunotherapy response in glioma” (Manuscript ID: medicina-2027221). Those comments are all valuable and very helpful for revising and improving our paper. We have replied to the comments carefully and made changes in the manuscript as suggested by the reviewer. In this revised version, changes to our manuscript were all highlighted within the document by using red colored text. The point-by-point response to the reviewers’ comments and concerns are as following:

Review#2

Evaluations

Comments:  

  1. the conclusion section in the manuscript could be better exhaustive.

Response: Thank you for your helpful comments.

We acknowledge that we did not provide an in-depth and detailed analysis in the conclusion section of our article, although I probably should have done so.

In response to your suggestion, we have reworked the conclusion section in the revised manuscript. Please see the conclusion section, page 19, lines 480-495.

Thanks!

  1. the manuscript is hard to read, maybe the collocations of figures should be different and to my opinion the discussione could less long.

Response: Thank you for your helpful comments.

We feel great thanks for your professional review work on our article. As you are concerned, the discussion part was indeed too long and does not highlight the views we want to express. According to your suggestion, we have made extensive modifications to the discussion section.We have cut out unnecessary sections (including results that were less necessary to discuss, and extended notes that were less relevant to this study). We have also simplified the language to make the ideas clearer and more concise. We believe that it is necessary and important for our study to review the studies conducted on model genes in glioma and other tumors and to explore the potential tumor-related functions of model genes in depth, so this section has not been overhauled. For more details, please see the Discussion section, page 16-18. As for figures, the combination of figures was determined after careful consideration. Each group of figures together showed the results for each step. We have enlarged the annotation font of the figures and added some text descriptions to the figures in the results section. We hope that these changes can help readers understand the results to be shown in the figures more easily.

Thanks!

We appreciate all your time and effort in helping us improve and clarify our manuscript. We also appreciate your extremely careful check for table and other related issues. We hope our revision has satisfactorily addressed the comments and will be acceptable for Medicina.

Sincerely,

Suxia Han, PhD.

Tel: 86-29-85324029. Fax: 86-29-85324029.

Email: shan87@xjtu.edu.cn

Reviewer 3 Report

Congratulations to Luo et al for a very well written and presented piece of work with extensive analysis and very interesting results. Please see below for some comments:

A. 

-Sample: the authors state that subjects studied were treated with immunotherapy but do not specify the type - it is not very clear to the reader whether all subjects received checkpoint inhibitors or other immunotherapy types, if any. Also, it is not clear what kinds of other modalities may have been used, such as radiotherapy. If such variations existed in the sample they should be included in the multivariable analysis. 

-similarly to above, was the ECOG PS (or other performance status scales) taken into consideration in the analysis?

-as above, is it known how many pts were treated/diagnosed with resection vs biopsy? if yes, was it included into the survival model The above questions, if not addressed statistically, should be included in the limitations section.

-limitations section: it is important to include any limitations of the study and future directions

B. Grammatical/writing comments

-in the methods/results sections there are sometimes short explanatory phrases, e.g. line 266 "The deletion of any of the MMR... microsatellite instability (MSI-H)" I would suggest to place such commentary in the Discussion.

-similarly to the above, citations should not be placed in the Results section

line 147: word "explore" is repeated twice

line 174: please do not start the paragraph with digits

Author Response

Dear Editor and the Reviewers,

We are grateful that Medicina us an opportunity to reconsider our revised manuscript titled “A novel immune gene-related prognostic score predicts survival and immunotherapy response in glioma” (Manuscript ID: medicina-2027221). Those comments are all valuable and very helpful for revising and improving our paper. We have replied to the comments carefully and made changes in the manuscript as suggested by the reviewer. In this revised version, changes to our manuscript were all highlighted within the document by using red colored text. The point-by-point response to the reviewers’ comments and concerns are as following:

Review#3

Evaluations

Comments:  

  1.  

-Sample: the authors state that subjects studied were treated with immunotherapy but do not specify the type - it is not very clear to the reader whether all subjects received checkpoint inhibitors or other immunotherapy types, if any. Also, it is not clear what kinds of other modalities may have been used, such as radiotherapy. If such variations existed in the sample they should be included in the multivariable analysis.

Response: Thank you for your helpful comments.

We apologize for not expressing clearly that the immune response predictive power of IGRPS model was indirectly estimated by the Tumor Immune Dysfunction and Exclusion (TIDE) algorithm, by which we compared the potential immune checkpoint efficacy of patients in high- and low-risk subgroups that were risk-stratified by IGRPS; patients in the high-risk group had a high TIDE prediction score, a high likelihood of immune evasion, and a poorer response to ICB treatment, and patients in the low-risk group were more likely to benefit from ICB treatment.

We acknowledge that this study lacks a cohort of glioma patients who have received immunotherapy to directly validate the predictive value of the model for immune response.We acknowledge that this study lacks a cohort of glioma patients who have received immunotherapy to directly validate the predictive value of the model for immune response. We hope that the explanations and improvements from the following two aspects can make up for this deficiency.

At present, immunotherapy has not been approved for glioma, and glioma immunotherapy is still in the pre-clinical and clinical research stages. The completed or ongoing clinical studies of glioma ICB immunotherapy are currently phase I / II clinical studies, so it is difficult to obtain the survival data of glioma patients that received ICB immunotherapy. This is also the reason why we did not include this indicator as an access standard in the research design。We have added this clarification in a new limitations section.

Tumor Immune Dysfunction and Exclusion (TIDE) algorithm is a computational method using gene expression profile to predict the ICB response and has a higher accuracy than PD-L1 expression level in predicting survival outcome of cancer patients treated with ICB Therapy(1-5). TIDE uses a set of gene expression markers to estimate 2 distinct mechanisms of tumor immune evasion, including dysfunction of tumor infiltration cytotoxic T lymphocytes (CTL) and exclusion of CTL by immunosuppressive factors(1). Patients with higher TIDE score have a higher chance of antitumor immune escape, thus exhibiting lower response rate of ICB treatment(1).Our model stratifies glioma patients by IGRPS risk, with low-risk subtypes having lower TIDE scores and more likely to benefit from immune checkpoint blockade therapies, reducing the genes to be sequenced to eight immune-related genes, thus significantly reducing the cost and burden to patients.

As stated in the introduction, part of the dilemma of glioma immunotherapy stems from the difficulty of accurately defining glioma patients with good ICB immune responses. We hope to screen glioma patients with good immune responses through our model, so that immunotherapy can obtain stable clinical benefits in such target patients, and thus shorten the distance between clinical research and clinical application of immunotherapy. It is hoped that the predictive value of immunotherapy efficacy of this model will contribute to the design of clinical immunotherapy research and promote the development of precise immunotherapy for glioma.

Thanks

-similarly to above, was the ECOG PS (or other performance status scales) taken into consideration in the analysis?

Response: Thank you for your helpful comments.

Good advice, but we are limited by the fact that the only clinical characteristics of gliomas we can obtain from TCGA database are age, gender, tumor grade, survival status and survival time. We hope to validate our model in the future with a larger sample of glioma patients with more clinical information, and to explore the relationship between the model and other clinical parameters to further explore the clinical application of the model. Thanks to your suggestion, we have added this regret in the limitations section.

-limitations section. it is important to include any limitations of the study and future directions

Response: Thank you for your helpful comments.

Thank you for your excellent suggestion, we add this paragraph at the end of the article. Please see page 18,lines 443-478.A limitation of the article is that the predictive power of the model for immunotherapy response was confirmed indirectly by the TIDE algorithm and was not validated using a cohort of patients who had received immunotherapy. We are actively collecting a cohort of glioma patients who have received immune checkpoint blockade and contain both survival information and transcriptional sequencing data, and hope to complete the validation of the model and further confirm its predictive performance in the future.

  1. Grammatical/writing comments

-in the methods/results sections there are sometimes short explanatory phrases, e.g. line 266 "The deletion of any of the MMR... microsatellite instability (MSI-H)" I would suggest to place such commentary in the Discussion.

Response: Thank you for your helpful comments.

We acknowledge that it is inappropriate to use too much explanation in the results section. Thanks to your suggestion, we have simplified and added this section to the methods section. The comparison of MMR gene expression levels between high and low risk subgroups is intended to further support the finding that patients in the low-risk group are more suitable for immunotherapy than those in the high risk group. In order to make readers better understand this concept, the function of the MMR genes has been briefly explained. We are not sure if you are comfortable with this change now, but of course if you still suggest to place those commentary in the Discussion, we will actively continue to make adjustments.

-similarly to the above, citations should not be placed in the Results section

Response: Thank you for your helpful comments.

Thanks to your suggestions, which have been very helpful in improving our article, we have adapted the Methods and Results sections.

line 147: word "explore" is repeated twice

Response: Thank you for your helpful comments.

We apologise for our carelessness. We have corrected it and we are very grateful to you for pointing it out.

line 174: please do not start the paragraph with digits

Response: Thank you for your helpful comments.

Thank you for your suggestion, we have adjusted it.

We appreciate all your time and effort in helping us improve and clarify our manuscript. We also appreciate your extremely careful check for table and other related issues. We hope our revision has satisfactorily addressed the comments and will be acceptable for Medicina.

Sincerely,

Suxia Han, PhD.

Tel: 86-29-85324029. Fax: 86-29-85324029.

Email: shan87@xjtu.edu.cn

Round 2

Reviewer 1 Report

The authors have answered the reviewer's suggestions and comments well. Since the authors asked for suggestion about Figure 7c, I personally feel the table format to be clearer and more appropriate. Oter than that the manuscript can be accepted in its current format.

Reviewer 3 Report

Congratulations to the authors on an improved manuscript. I believe the changes are satisfactory for publication in this journal.